# A nanocompartment system contributes to defense against oxidative stress in *Mycobacterium tuberculosis*

Katie A Lien[1†], Kayla Dinshaw[1†], Robert J Nichols[2], Caleb Cassidy-Amstutz[2], Matthew Knight[3], Rahul Singh[4], Lindsay D Eltis[4], David F Savage[2], Sarah A Stanley[1,5]*

[1]Department of Molecular and Cell Biology, Division of Immunology and Pathogenesis, University of California, Berkeley, Berkeley, United States; [2]Department of Molecular and Cell Biology, Division of Biochemistry, Biophysics and Structural Biology, University of California, Berkeley, Berkeley, United States; [3]Department of Plant and Microbial Biology, University of California, Berkeley, Berkeley, United States; [4]Department of Microbiology and Immunology, The University of British Columbia, Vancouver, Canada; [5]School of Public Health, Division of Infectious Diseases and Vaccinology, University of California, Berkeley, Berkeley, United States

**Abstract** Encapsulin nanocompartments are an emerging class of prokaryotic protein-based organelle consisting of an encapsulin protein shell that encloses a protein cargo. Genes encoding nanocompartments are widespread in bacteria and archaea, and recent works have characterized the biochemical function of several cargo enzymes. However, the importance of these organelles to host physiology is poorly understood. Here, we report that the human pathogen *Mycobacterium tuberculosis* (Mtb) produces a nanocompartment that contains the dye-decolorizing peroxidase DyP. We show that this nanocompartment is important for the ability of Mtb to resist oxidative stress in low pH environments, including during infection of host cells and upon treatment with a clinically relevant antibiotic. Our findings are the first to implicate a nanocompartment in bacterial pathogenesis and reveal a new mechanism that Mtb uses to combat oxidative stress.

*For correspondence: sastanley@berkeley.edu

†These authors contributed equally to this work

Competing interest: The authors declare that no competing interests exist.

## Editor's evaluation

The authors identify a two-gene operon encoding for a peroxidase and an encapsulin in *M. tuberculosis*. They demonstrate that this system is important for *M. tuberculosis* to resist oxidative stress at acidic pH in vitro and in vivo in infected macrophages. The work comprises a significant advance in the understanding of mycobacterial pathogenicity and the role of bacterial encapsulins.

## Introduction

The success of *Mycobacterium tuberculosis* (Mtb) as a human pathogen results from its ability to avoid, resist, or subvert immunological mechanisms that are effective against most pathogens. Mtb has a remarkable ability to withstand the microbicidal arsenal of host macrophages, including the low pH of the lysosome and a diverse array of toxic reactive oxygen species (ROS) produced by the host respiratory burst. However, relatively little is known about the molecular mechanisms of resistance to these toxic insults.

Upon phagocytosis of a pathogen, macrophages and neutrophils unleash a potent microbicidal arsenal, including a respiratory burst that leads to the rapid release of ROS that kill and degrade

most microbes (*El-Benna et al., 2009*; *Nathan, 2006*; *Segal, 2005*). The production of ROS is initiated when NADPH oxidase assembles on pathogen-containing phagosomes and initiates a series of reactions that produce superoxide, hydrogen peroxide, hyperchlorous acid, lipid peroxides, and other toxic reactive molecules. Importantly, Mtb is highly resistant to oxidative stress, enabling the bacterium to survive within macrophages and neutrophils and establish chronic infection (*Chan et al., 1992*). Although some oxidative defense mechanisms have been characterized (*Ehrt and Schnappinger, 2009*; *Nambi et al., 2015*; *Voskuil et al., 2011*), the Mtb genome encodes a large number of enzymes that could participate in defense against oxidative stress that remain uncharacterized. For example, Mtb encodes ~15 peroxidases, a class of enzymes with diverse functions that form the front line of oxidative defense in bacteria (*Mishra and Imlay, 2012*; *Singh and Eltis, 2015*). It is unclear why Mtb encodes such a diversity of peroxidases, and little is known about what individual roles these enzymes may play in both resistance to oxidative stress and in normal bacterial physiology. One possibility is that specific peroxidase genes may participate in defense against different types of ROS generated during the respiratory burst. One of the most intriguing peroxidases encoded by Mtb is DyP, a member of the dye-decolorizing peroxidase family of proteins. DyP enzymes are unique amongst peroxidases in that some family members have been shown to have optimal activity at pH 4–5 (*Brown et al., 2012*; *Singh and Eltis, 2015*; *Uchida et al., 2015*). Furthermore, DyP is the only Mtb peroxidase predicted to be packaged inside a proteinaceous organelle known as a bacterial nanocompartment (*Contreras et al., 2014*).

Bacterial cells were long thought to lack compartmentalization of function. However, the identification of organelle-like structures including microcompartments, anammoxosomes, magnetosomes, and, most recently, encapsulin nanocompartments has revolutionized our understanding of bacterial cell biology (*Ferousi et al., 2017*; *Kerfeld et al., 2018*; *Uebe and Schüler, 2016*). Characterized nanocompartments are proteinaceous shells that are 24–45 nm in diameter and comprised of 60–240 subunits of a single protomer (*Giessen et al., 2019*; *Nichols et al., 2017*). This shell surrounds ('encapsulates') an enzymatic cargo protein (*Nichols et al., 2017*). Although putative encapsulin systems have been identified in >900 bacterial and archaeal genomes (*Giessen and Silver, 2017*; *Nichols et al., 2020*), very little is known about their physiological function. Based on genomic organization, encapsulin systems are often predicted to compartmentalize enzymes involved in oxidative stress defense, iron storage, and anaerobic ammonium oxidation (*Giessen and Silver, 2017*). However, the function of an encapsulin system has only ever been demonstrated in a single bacterial species, *Myxococcus xanthus* (*McHugh et al., 2014*), and the physiological role of nanocompartments is largely unexplored.

Here, we demonstrate that an Mtb nanocompartment containing DyP is crucial for resisting oxidative stress at low pH. Mutants lacking the nanocompartment are highly attenuated when exposed to $H_2O_2$ at pH 4.5, the pH of the lysosome. We show that this attenuation is linked to fatty acid metabolism in the bacteria. Further, we show that encapsulation of the DyP enzyme promotes its function, the first report to demonstrate a role for encapsulation in endogenous bacterial physiology. Finally, we show that the Mtb nanocompartment system is required for full growth in macrophages, and for resistance to the antibiotic pyrazinamide (PZA), demonstrating the importance of this system for a globally significant pathogen.

## Results

The Mtb genome encodes the predicted encapsulin gene *Rv0798c*/Cfp29 (*Contreras et al., 2014*) in a two-gene operon with *Rv0799c*, the dye-decolorizing peroxidase DyP (*Figure 1A*). Overexpression of the predicted Mtb encapsulin gene in *Escherichia coli* was previously shown to result in the formation of nanocompartment-like structures (*Contreras et al., 2014*). Three potential cargo proteins for the nanocompartment were proposed based on a putative shared encapsulation targeting sequence: DyP, FolB, and BrfB. In *E. coli*, overexpression of each protein with Cfp29 resulted in encapsulation. However, this study did not address whether Mtb produces endogenous nanocompartments or identify the specific function of these compartments in Mtb biology. A transposon screen has identified Cfp29 as a gene required for growth in mice (*Zhang et al., 2013*) and Cfp29 has long been known as an immunodominant T cell antigen in both mice and human TB patients (*Weldingh and Andersen, 1999*). Taken together, these results suggest that Mtb may produce an encapsulin nanocompartment that is important for pathogenesis.

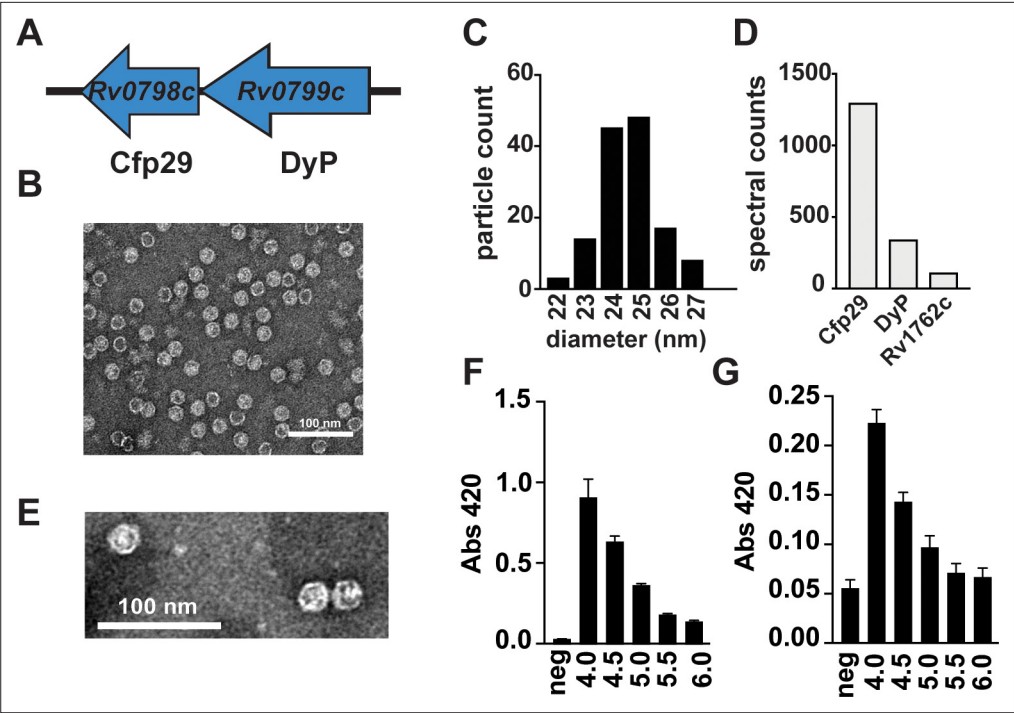

**Figure 1.** Mtb produces endogenous nanocompartments that package a peroxidase. (**A**) Schematic of the nanocompartment operon in Mtb that encodes the encapsulin shell protein (Cfp29) and the dye-decoloring peroxidase cargo protein (DyP). (**B**) Transmission electron microscopy (TEM) of Cfp29 encapsulin proteins purified following heterologous expression of the Mtb nanocompartment operon in *E. coli*. (**C**) Size distribution of Cfp29 protomers purified from *E. coli*. (**D**) Peptide counts from mass spectrometry analysis of endogenous nanocompartments purified from Mtb. (**E**) TEM of endogenous nanocompartments purified from Mtb. Peroxidase activity of (**F**) unencapsulated and (**G**) encapsulated DyP (5 nM) using ABTS (480 nM) as a substrate in the presence of $H_2O_2$ (480 nM) at varying pH levels (4.0–6.0) as reported by a change in the absorbance at 420 nm. neg, no added enzyme.

The online version of this article includes the following figure supplement(s) for figure 1:

**Figure supplement 1.** Nanocompartment purification and complementation strategies.

To confirm the previous finding that heterologous expression of *Rv0798c* and *Rv0799c* in a host species results in the assembly of an encapsulin system, we expressed these genes in *E. coli* and isolated nanocompartments. Clarified protein lysates from *E. coli* were purified by ultracentrifugation and size-exclusion chromatography. Assembled encapsulin nanocompartments are distinguishable by their high molecular weight (*Nichols et al., 2017*). Indeed, a fraction from the purification contained a high molecular weight species > 260 kDa observable on an SDS-PAGE gel (*Figure 1—figure supplement 1A*). Fractions containing putative nanocompartments were pooled and imaged using transmission electron microscopy, which revealed the presence of icosahedral structures with the expected diameter of ~25 nm (*Figure 1B and C*).

To determine whether Mtb produces nanocompartments under normal laboratory growth conditions, we performed an ultracentrifugation-based nanocompartment isolation strategy using wild-type H37Rv strain bacteria grown to mid-log phase (*Figure 1—figure supplement 1B*). Mass spectrometry analysis of the nanocompartment fraction identified both the encapsulin protein Cfp29 and the peroxidase DyP (*Figure 1D*). TEM analysis confirmed the presence of nanocompartment particles ~25 nm in diameter (*Figure 1E*). Interestingly, Rv1762c, a protein of unknown function, was consistently identified in purified nanocompartment preparations from Mtb (*Figure 1D*). We were unable to identify either FolB or BrfB in nanocompartments from Mtb, suggesting that these proteins are not endogenous substrates for encapsulation under normal laboratory growth conditions for Mtb.

Cfp29 ('culture filtrate protein 29') was originally identified in the supernatants of Mtb cells grown in axenic culture (*Rosenkrands et al., 1998*). As Cfp29 lacks a secretory signal sequence and is part

of a large macromolecular complex, it is unclear how a nanocompartment could be actively secreted. However, nanocompartment structures are remarkably stable and it is possible that nanocompartments released from dying bacteria accumulate in culture as they are highly resistant to proteolysis/degradation (*Cassidy-Amstutz et al., 2016*; *Nichols et al., 2017*).

DyP proteins are known to have low pH optima (*Contreras et al., 2014*; *Uchida et al., 2015*). To demonstrate that Mtb DyP has a similarly low pH optimum, SUMO-tagged unencapsulated DyP was purified from *E. coli* (*Figure 1—figure supplement 1C*), and the peroxidase activity was evaluated at a range of pH values using the ABTS (2,2'-azino-bis(3-ethylbenzothiazoline-6-sulfonic acid)) dye-decolorizing assay previously used to characterize DyP proteins (*Ahmad et al., 2011*; *Contreras et al., 2014*). Similar to the *Vibrio cholerae* DyP (*Uchida et al., 2015*), purified Mtb DyP had increased enzymatic activity in low pH environments, with the greatest efficacy at pH 4.0, the lowest pH tested (*Figure 1F*). We next tested the ability of encapsulated DyP to degrade ABTS across a range of pH values. Similar to free DyP, the encapsulated enzyme had the highest activity at pH ~4.0 (*Figure 1G*). Taken together, these data demonstrate the stability and functionality of Mtb nanocompartments under acid stress, a condition that mimics the host lysosomal environment. The Mtb DyP protein was previously shown to function as a bona fide dye-decolorizing peroxidase (*Contreras et al., 2014*). Because peroxidases consume $H_2O_2$, they often participate in defense against oxidative stress (*Mishra and Imlay, 2012*). During macrophage infection, the phagocytes initiate an oxidative burst that exposes Mtb to $H_2O_2$ (*El-Benna et al., 2009*). We therefore reasoned that DyP-containing nanocompartments may function to protect Mtb from $H_2O_2$-induced stress. To test this hypothesis, we created a mutant strain lacking both genes from the nanocompartment operon (*Figure 1A*, Δoperon). Lysates from Δoperon mutants were used for nanocompartment purification (data not shown) and western blot analysis using an antibody for Cfp29 as a probe (*Figure 1—figure supplement 1D*). Data from these analyses revealed that Δoperon mutants cannot produce viable nanocompartments. In addition, we isolated a transposon mutant (*DyP::Tn*) containing an insertion in *DyP*, a mutation that eliminated expression of both Cfp29 and DyP (*Figure 1—figure supplement 1E*).

To test whether DyP-containing nanocompartments are required for defense against $H_2O_2$, we exposed wild-type and *DyP::Tn* Mtb to increasing concentrations of $H_2O_2$ and monitored bacterial survival. Mutants lacking DyP nanocompartments were more susceptible to $H_2O_2$ when compared with wild-type bacteria as measured by $OD_{600}$ (*Figure 2A*) and by plating for colony-forming units (CFU) (*Figure 2B*). However, the phenotype was relatively modest. We next reasoned that DyP nanocompartments might be required to resist oxidative stress in acidic environments that more closely mimic the in vivo environment. During infection, Mtb bacilli encounter the low pH of the phagolysosome and the ability to tolerate low pH is required for Mtb survival in both infected macrophages and mice (*Vandal et al., 2008*). We therefore tested the susceptibility of DyP mutants to a combination of $H_2O_2$ and acid stress (pH 4.5). Wild-type Mtb was able to withstand these conditions and did not significantly decrease in number over a 3-day exposure period (*Figure 2C*). In contrast, the Δoperon mutant was resistant to each stressor individually, but was highly susceptible to $H_2O_2$ at pH 4.5 (*Figure 2C*). Thus, DyP-containing nanocompartments are required to protect bacteria from oxidative stress at low pH.

We next sought to determine whether encapsulation of DyP is important for its activity in bacterial cells. To accomplish this, *DyP::Tn* mutants were complemented with constructs expressing DyP alone (pDyP), Cfp29 encapsulin alone (pCfp29), or both proteins (pOperon). As expected, expression of DyP did not restore production of nanocompartments, whereas expression of Cfp29 alone resulted in the formation of empty nanocompartment structures (*Figure 1—figure supplement 1E*). Coexpression of both proteins resulted in the formation of DyP-containing nanocompartments in the *DyP::Tn* mutant (*Figure 1—figure supplement 1E*). We exposed the full set of complemented mutants to $H_2O_2$ at pH 4.5 and determined survival after 3 days by plating for CFU. As expected, *DyP::Tn* was attenuated for survival under these conditions when compared to wild-type bacteria (*Figure 2D*). Restoring expression of the nanocompartment shell protein in the absence of DyP had no effect on bacterial survival (*Figure 2D*). Complementation by overexpression of unencapsulated DyP was sufficient to confer almost wild-type levels of resistance to oxidative and acid stress (*Figure 2D*). However, complementation with both Cfp29 and DyP resulted in enhanced resistance to these stressors when compared to mutants complemented with DyP alone (*Figure 2D*), suggesting that encapsulation of DyP enhances its function.

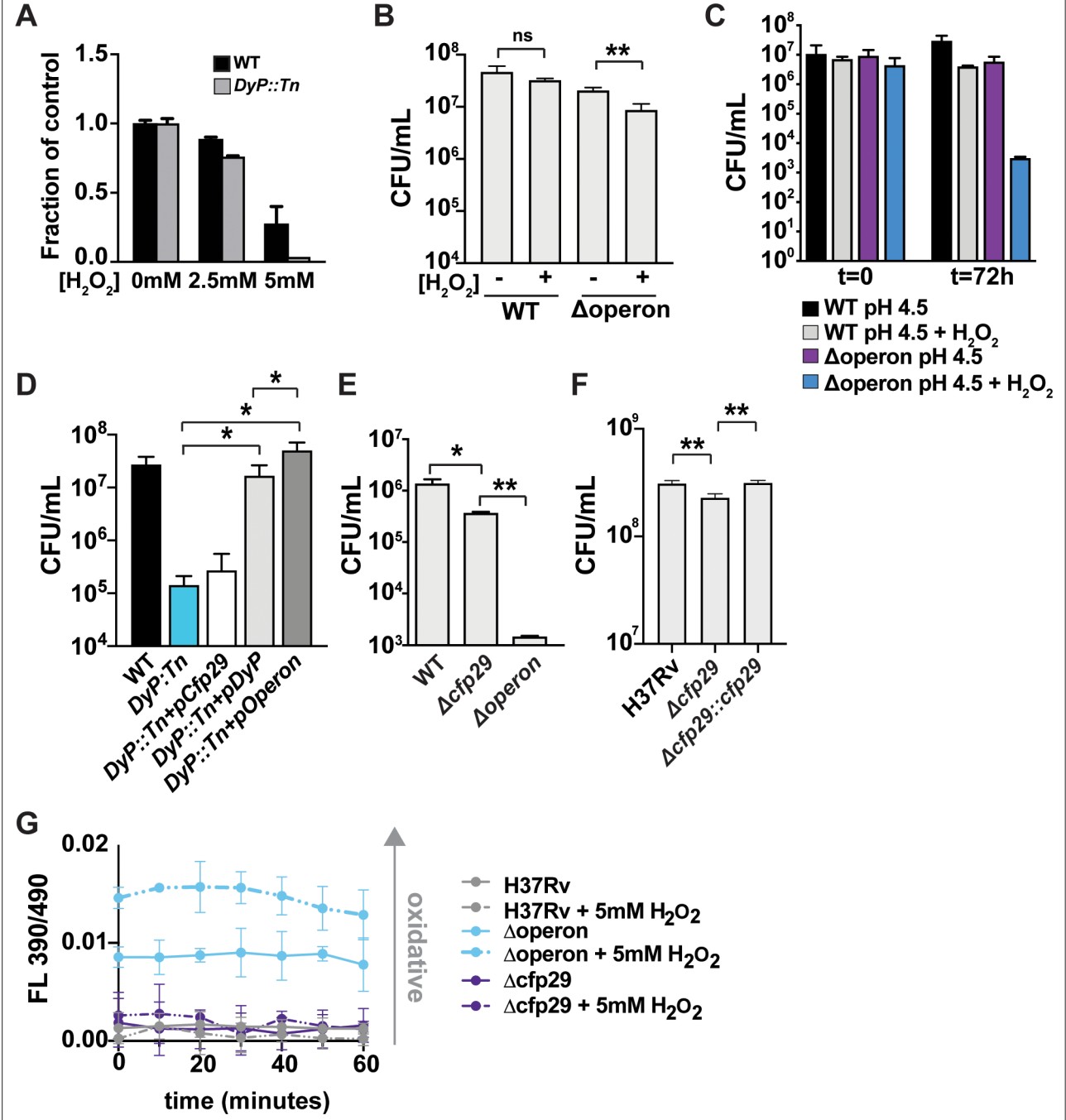

**Figure 2.** Nanocompartments protect Mtb from oxidative stress in acidic environments. (**A**) OD$_{600}$ measurements of wild-type and *DyP::Tn* Mtb grown in 7H9 medium following exposure to H$_2$O$_2$ for 96 hr. Values reported are normalized to the untreated controls. CFU enumeration of wild-type and Mtb nanocompartment mutants grown in (**B**) standard 7H9 medium (pH 6.5) and (**C–F**) acidified 7H9 medium (pH 4.5) following exposure to oxidative stress (2.5 mM H$_2$O$_2$) for 72 hr. (**G**) Fluorescence emissions of wild-type and Δoperon Mtb expressing mrx1-roGFP exposed to 5 mM H$_2$O$_2$ at pH 4.5 in 7H9 medium for 60 min. Data are reported as a ratio of fluorescence emissions following excitation at 490 nm and 390 nm. Figures are representative of at least two (**E, F**) or three (**A–D, G**) independent experiments. p-Values were determined using unpaired t-test. *p<0.05, **p<0.01.

In the mutant complemented with DyP alone, it is possible that overexpression of the peroxidase compensates for the lack of encapsulation. We therefore examined the importance of DyP encapsulation in mutant strains in which we deleted *Cfp29* only (Δcfp29). Deletion of *Cfp29* resulted in an ~0.5 log decrease in bacterial viability after 3 days of exposure to H$_2$O$_2$ at pH 4.5, confirming that encapsulation of the peroxidase in the shell protein is important for full protection (***Figure 2E***). As

expected, greater attenuation was observed in the mutant lacking both DyP and Cfp29 (Δoperon). Complementation of the Δcfp29 mutant fully restored resistance (*Figure 2F*). Taken together, these data demonstrate that encapsulation of DyP has a modest, yet significant impact on protection against oxidative stress in low pH conditions.

We next sought to test whether nanocompartment mutants have altered redox homeostasis in the presence of oxidative stress. We transformed wild-type and mutant strains with Mrx1-roGFP, a fluorescent reporter of redox potential in mycobacteria (*Bhaskar et al., 2014*). Remarkably, the Δoperon mutant had an intracellular environment at baseline that was more oxidizing than wild-type bacteria exposed to $H_2O_2$ (*Figure 2G*), indicating a failure of redox homeostasis in mutants lacking the DyP encapsulin system. This oxidizing cellular environment was further enhanced by treatment with $H_2O_2$ (*Figure 2G*). We did not observe a difference in the redox state of Δcfp29. Thus, mutants lacking DyP-containing nanocompartments exhibit significant dysregulation of redox homeostasis.

The Mtb genome encodes ~15 putative peroxidase genes, including DyP. To determine whether other bacterial peroxidases participate in the defense against oxidative stress at low pH, we performed a transposon sequencing (Tn-seq) screen (*Griffin et al., 2012*; *van Opijnen and Camilli, 2013*). To do so, we created a transposon library containing ~100K individual transposon mutants and exposed

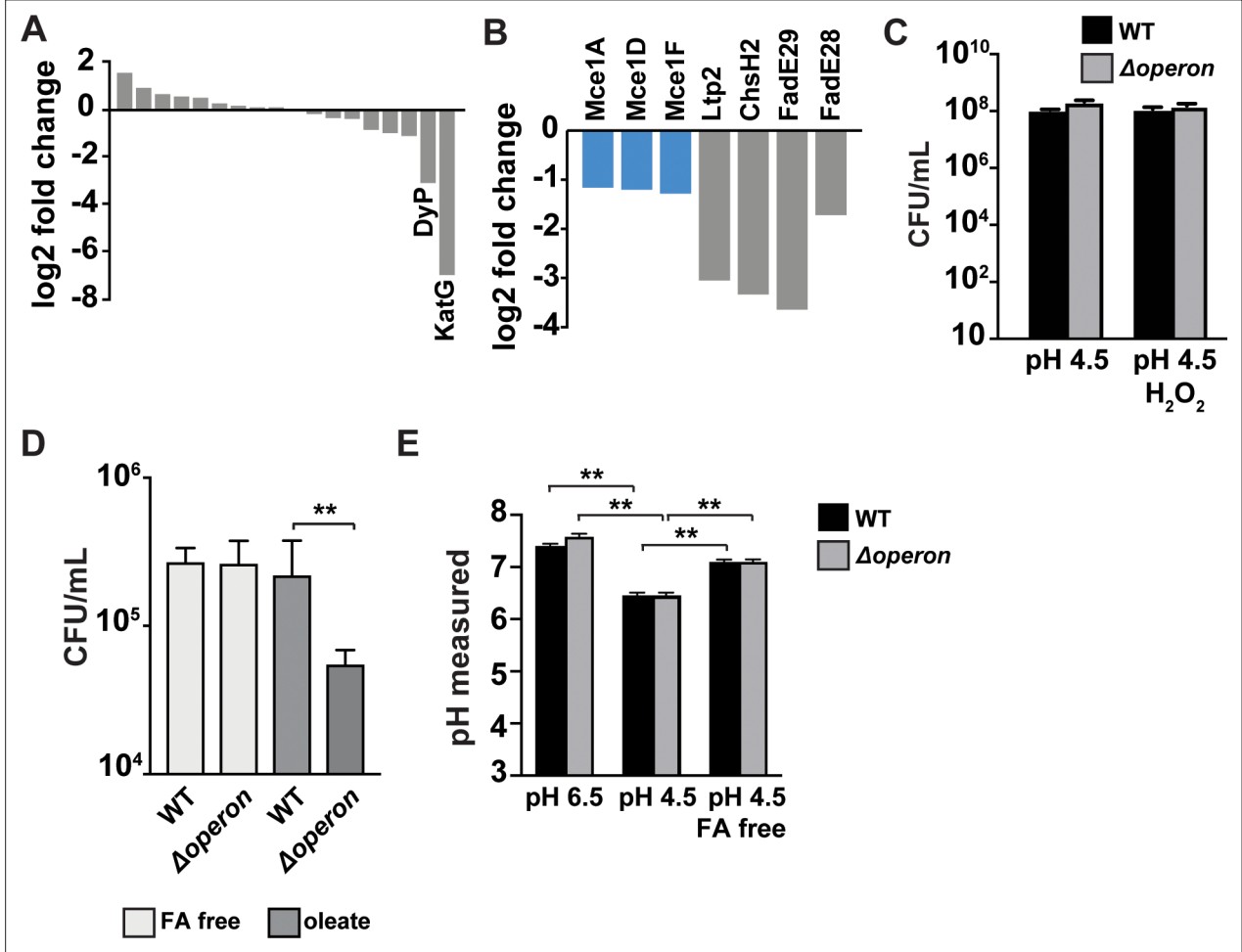

**Figure 3.** Susceptibility of Mtb nanocompartment mutants to oxidative and acid stress is mediated by free fatty acids. (**A**) Transposon sequencing (Tn-seq) data showing normalized sequence reads per gene for all putative Mtb peroxidases, catalases, and superoxide dismutases and (**B**) lipid and cholesterol metabolism Mtb mutants that were significantly attenuated following 72 hr exposure to 2.5 mM $H_2O_2$ at pH 4.5. (**C**) CFU enumeration of wild-type Mtb and Δoperon mutants following 24 hr exposure to 2.5 mM $H_2O_2$ at pH 4.5 in Sauton's minimal medium and (**D**) 72 hr exposure to 2.5 mM $H_2O_2$ at pH 4.5 in 7H9 medium prepared using fatty acid (FA)-free bovine serum albumin (BSA) ± oleic acid (150 μM). (**E**) Intrabacterial pH measurements of wild-type and Δoperon Mtb expressing pUV15-pHGFP following 20 min exposure to 5 mM $H_2O_2$ at pH 6.5 or pH 4.5. 7H9 medium was prepared with standard BSA or FA-free BSA. Figures are representative of at least two (**D**) or three (**A–C, E**) independent experiments. p-Values were determined using an unpaired t-test. *p<0.05, **p<0.01.

this library to $H_2O_2$ at pH 4.5 for 3 days, at which point the surviving bacteria were plated. Transposon gene junctions were amplified and sequenced from the recovered bacteria, and the sequencing data were analyzed using TRANSIT (*DeJesus et al., 2015*; *Supplementary file 1*). These data revealed that of the 18 putative enzymes encoded by Mtb that could be involved in oxidative defense, including peroxidases, catalases, and superoxide dismutases, only two genes were required for survival in the culture conditions—*DyP* and *KatG* (*Figure 3A*). *KatG* encodes a catalase-peroxidase that is important for defense against oxidative stress in the host (*Li et al., 1998*; *Pym et al., 2002*). Coincidentally, KatG is also required for activation of isoniazid (INH), a central component of anti-TB therapy. Approximately 10% of TB cases are caused by INH-resistant bacteria, many of which have loss-of-function mutations in KatG (*Seifert et al., 2015*). Importantly, KatG variants that do not activate INH often display a concomitant decrease in peroxidase and catalase activity (*DeVito and Morris, 2003*; *Mo et al., 2004*; *Wei et al., 2003*). Both the high proportion of KatG mutant bacteria in the human population and studies of virulence in animals suggest that KatG mutants are still capable of growth in vivo (*Nieto et al., 2016*; *Pym et al., 2002*). The fact that both DyP and KatG mutants are important for survival in oxidative stress at low pH suggests that the DyP nanocompartment may provide redundancy that compensates for a loss of KatG in INH-resistant strains of Mtb.

In the Tn-seq data, we observed that mutants involved in lipid or cholesterol metabolism were also attenuated when exposed to $H_2O_2$ at pH 4.5 (*Figure 3B*). Mtb biology is highly linked to lipid biology; Mtb granulomas are rich in free fatty acids (*Kim et al., 2010*) and the bacteria utilize lipids as a source of nutrition both during macrophage infection ex vivo and growth in vivo (*Marrero et al., 2010*; *Russell, 2011*; *Russell et al., 2019*). In particular, mutants containing transposon insertions in three independent subunits of the Mce1 complex, which is required for import of fatty acids (*Nazarova et al., 2019*), were susceptible to oxidative stress under acidic conditions (*Figure 3B*). The standard Mtb culture medium, 7H9, contains bovine serum albumin (BSA), which has binding sites for fatty acids and may also serve as a cholesterol shuttle in serum (*Sankaranarayanan et al., 2013*; *van der Vusse, 2009*). We hypothesized that fatty acids carried by albumin may become toxic following exposure to oxidative stress and low pH and that import of fatty acids into the bacteria might be a protective mechanism. . We therefore tested whether fatty acids are required for the susceptibility of Δoperon mutants using exposure to acid and oxidative stress in Sauton's broth, a minimal medium that lacks BSA, and found that in the absence of fatty acids the Δoperon mutants persisted to the same degree as wild-type bacteria (*Figure 3C*). To test whether lipids bound to BSA mediate the susceptibility of DyP mutants to acid and oxidative stress, we cultured wild-type and Δoperon Mtb in media constituted with fatty acid-free BSA and exposed the bacteria to $H_2O_2$ at pH 4.5 for 3 days. In the absence of BSA-bound lipids, we found that the Δoperon mutant was not susceptible to oxidative stress at low pH (*Figure 3D*). 7H9 medium prepared with lipid-free BSA was then reconstituted with oleic acid, an abundant lipid found in mammalian systems (*Abdelmagid et al., 2015*). Reconstitution of fatty acid-free medium with oleic acid restored toxicity to nanocompartment mutants under conditions of acid and oxidative stress (*Figure 3D*). Taken together, these data suggest that the phagosomal lipids used by Mtb as a carbon source may be toxic to bacteria lacking functional nanocompartments under conditions of oxidative stress.

We considered the possibility that lipids present in the medium might disrupt Mtb membranes, resulting in altered pH homeostasis and a drop in cytosolic pH. Interestingly, in acidified medium we found that the intracellular pH of wild-type bacteria dropped from ~pH 7.5 to ~pH 6.2, and this decrease was dependent on the presence of albumin-bound fatty acids (*Figure 3E*). These data confirm published results demonstrating that the cytosolic pH of Mtb drops significantly during acid exposure (*Vandal et al., 2008*). A lower cytosolic pH would possibly impair the function of many other bacterial enzymes important for resisting oxidative stress and place increasing importance on enzymes that can function in acidic environments, such as DyP (*Uchida et al., 2015*). Taken together, these data may explain why functional nanocompartments are critical for bacterial survival when Mtb is exposed to acid and oxidative stress in a fatty acid-rich environment.

Our in vitro findings that DyP nanocompartments are important for defense against a combination of oxidative stress, low pH, and fatty acids suggested that this system may be important for Mtb survival in the phagosomal environment. Therefore, we sought to determine whether DyP nanocompartments are required for bacterial growth in host cells. We found that the Δoperon strain was consistently attenuated for growth in murine bone marrow-derived macrophages (*Figure 4A*). We

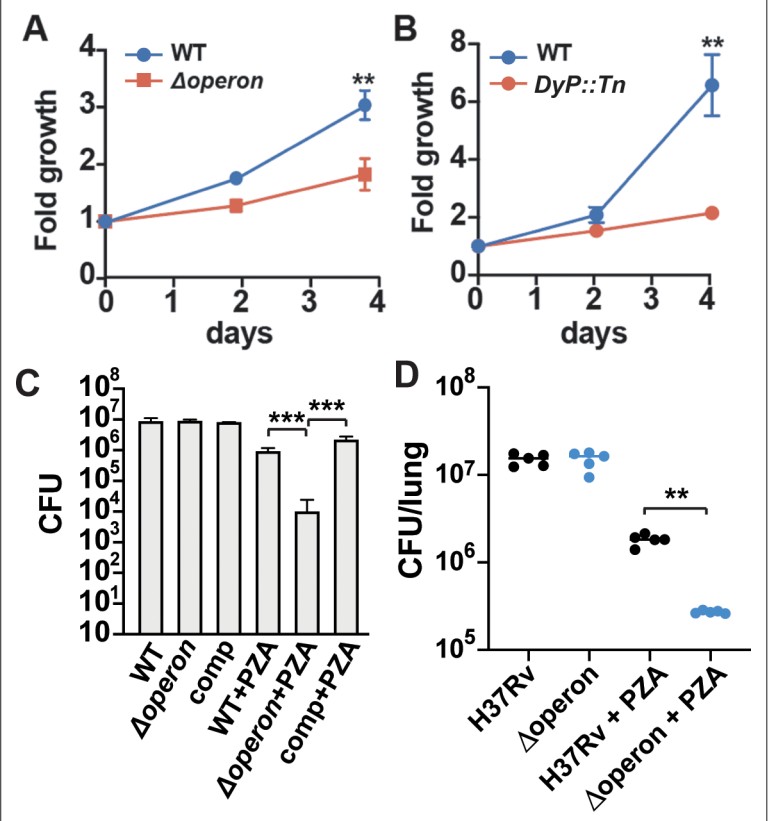

**Figure 4.** Nanocompartment mutants are attenuated for survival in macrophages and are more susceptible to pyrazinamide (PZA) treatment. CFU enumeration of wild-type Mtb and (**A**) Δoperon or (**B**) DyP::Tn mutants during infection of murine bone marrow-derived macrophages. Macrophages were infected with a bacterial MOI of 1, and CFUs were enumerated immediately following phagocytosis and at days 2 and 4. Error bars are SD from four replicate wells. (**C**) CFU enumeration of wild-type and Mtb Δoperon mutants following 72 hr exposure to PZA (24 µg/mL) and $H_2O_2$ (2.5 mM) in acidified 7H9 medium (pH 5.5). Comp = Δoperon + pOperon. (**D**) Infection of BALB/C mice with WT and Δoperon mutant with and without treatment with 150 mg/kg PZA. CFU at 35 days post infection in the lung is shown. (**A**) and (**B**) are representative of at least five independent experiments; (**C**) is representative of three experiments and (**D**) is representative of two experiments. p-Values were determined using an unpaired t-test. **p<0.01; ***p<0.001.

attempted to complement this phenotype; however, the limited dynamic range between wild-type and mutant in macrophages made clean interpretation of the complementation results difficult. As these genes are predicted to be in a two-gene operon with no other predicted genes in close proximity, polar effects are extremely unlikely. However, to confirm that the phenotype of the mutants in macrophages is due to the loss of the two-gene encapsulin operon, we repeated the experiment with the independently derived *Dyp:Tn* mutant and also observed attenuated growth in macrophages across repeated experiments (*Figure 4B*).

One of the antibiotics used to treat TB infection, PZA, is most effective in low pH environments, such as the phagolysosome. PZA is thought to disrupt the cell wall of Mtb, which leads to acidification of the bacterial cytosol. A recent study in human TB patients demonstrated that the pH of necrotic lung cavities is ~5.5, a finding that may also explain the efficacy of PZA in treating Mtb infection (*Kempker et al., 2017*). Since DyP is critical for protection of Mtb against oxidative stress at low pH, we considered whether nanocompartments may mediate Mtb resistance to PZA. To test bacterial susceptibility to PZA, we treated wild-type and Δoperon Mtb with PZA in the presence of $H_2O_2$ at pH 5.5. We found that mutants lacking the DyP nanocompartment were sensitive to concentrations of PZA that did not impact growth of the wild-type bacteria (*Figure 4C*). We next set out to test whether mutants lacking the DyP operon are susceptible to PZA treatment in vivo. We infected BALB/C mice with wild-type and mutant bacteria by the aerosol route with a dose of ~250 bacteria instilled on day 0 of infection.

PZA treatment began on day 14 and continued until day 35, at which time we enumerated CFU in the lungs. We did not find that Δoperon mutants were more attenuated than wild-type bacteria in mouse lungs at 35 days post infection, at least under the conditions examined. However, Δoperon mutants were significantly more susceptible to treatment with PZA in vivo (*Figure 4D*), demonstrating that the DyP encapsulin operon is required for endogenous resistance to PZA treatment.

## Discussion

Here, we report that a protein nanocompartment in Mtb encapsulates the dye-decolorizing peroxidase DyP. By demonstrating that the encapsulin system is important for resisting oxidative stress at low pH, we provide the first evidence that encapsulin systems contribute to oxidative stress in a bacterial pathogen, and only the second demonstration of a functional encapsulin system in any species. Furthermore, we expand our understanding of the importance of encapsulation in enzyme function by demonstrating that the protein shell encapsulating DyP enhances its function in vivo. Finally, we demonstrate that the DyP nanocompartment system is essential for the ability of Mtb to grow in host macrophages and to resist killing by the antibiotic PZA. These findings are the first demonstration that a nanocompartment system is important for virulence in any bacterial pathogen.

Bacterial protein-based compartments are emerging as functional equivalents of eukaryotic organelles that can sequester enzymatic reactions from the environment of the cytosol. Bacteria appear to have two predominant classes of protein based organelles, the relatively well-understood bacterial microcompartment (*Kerfeld et al., 2018*) and the more recently discovered bacterial nanocompartment (*Nichols et al., 2017*). Rather than using a lipid membrane for compartmentalization, both bacterial microcompartments and nanocompartments consist of a porous protein shell that surrounds enzymatic proteins. Bacterial microcompartments that enable utilization of specialized nutrients including ethanolamine and 1,2 propane diol have previously been implicated in pathogenesis of *E. coli* and *Salmonella enterica,* respectively (*Jakobson and Tullman-Ercek, 2016*). Nanocompartments, which are 10 times smaller than a typical bacterial microcompartment, have been identified bioinformatically in over 900 bacterial species; however, very few studies have addressed the role that nanocompartments play in bacterial physiology. Bioinformatic analysis and the heterologous expression of nanocompartment systems from diverse bacteria in *E. coli* have suggested that these systems might function in iron mineralization, oxidative and nitrosative stress resistance, and anaerobic ammonium oxidation (*Giessen and Silver, 2017*). However, the only nanocompartment that has been shown to be important for survival under conditions of stress is the ferritin-like protein containing compartment in *M. xanthus*, which is required for resistance to $H_2O_2$ (*McHugh et al., 2014*). Our demonstration that an encapsulin system is required for resistance to oxidative stress in Mtb significantly broadens the potential role and impact of nanocompartment systems in bacterial physiology.

A previous report examined the Mtb nanocompartment by heterologous expression of the encapsulin gene in *E. coli*. Coexpression with three potential substrates identified bioinformatically based on a putative signal sequence suggested that three proteins can be encapsulated by the Mtb nanocompartment: DyP, FolB, and BrfB (*Contreras et al., 2014*). Here, we show for the first time that Mtb produces this nanocompartment. However, of the three potential cargo proteins, only DyP was identified in encapsulin systems purified from logarithmically growing bacteria. Why is the DyP protein encapsulated? There is almost no understanding of the role of encapsulation in bacterial physiology despite the widespread presence of encapsulins across bacterial species. In this study, we found that DypB alone can protect the bacteria from oxidative stress at low pH. It is possible that in some circumstances, particularly when the stressor is not maximal, encapsulation of DyP may not be necessary. We expect that further investigation of this nanocompartment system in Mtb and in other bacterial species will reveal additional roles of encapsulins in the bacterial physiology.

We showed that Mtb mutants lacking the DyP encapsulin system have an altered redox balance at baseline, with a significantly more oxidative cytosol than wild-type. However, under standard growth conditions, the mutants are not defective. While mutants exhibit a small defect for survival when exposed to $H_2O_2$ at pH 6.5, they are highly attenuated at low pH in the presence of $H_2O_2$. Interestingly, the attenuation of the mutants under these conditions requires the presence of fatty acids bound to BSA in the culture medium. Why is the DyP encapsulin system required for growth under these conditions? It is possible that this system is required for detoxifying an oxidative species other than $H_2O_2$ itself, such as a lipid peroxide. It is also possible that the low pH compromises the integrity of

the cell wall such that oxidative damage is sustained, or that the low pH magnifies the importance of DyP, which has a low pH optimum. Nevertheless, although the exact function of the DyP encapsulin system remains to be elucidated, this system is clearly important for pathogenesis as mutants fail to grow in host macrophages. Intriguingly, encapsulin systems have been shown to confer stability to the enzymatic cargo under harsh conditions, including heat and protease exposure (*Nichols et al., 2017*). Cfp29 was originally identified as a component of short-term culture filtrate present in broth of growing cells, leading to the supposition that this protein is secreted (*Rosenkrands et al., 1998*). Given that nanocompartments spontaneously assemble inside cells (*Cassidy-Amstutz et al., 2016*; *Nichols et al., 2017*; *Snijder et al., 2016*), it is unlikely that a structure the size of a ribosome could be secreted. We speculate that nanocompartments are released from dying cells, and that their extreme stability promotes their accumulation in axenic bacterial culture. Indeed, this stability may partially explain why Cfp29 is a known immunodominant antigen for T cells (*Rosenkrands et al., 1998*). Taken together, these data suggest the intriguing possibility that nanocompartments released from dying cells in vivo could continue to provide defense against oxidative stress for the surviving bacteria.

We found that only two peroxidases, KatG and DyP, are required for survival in the presence of $H_2O_2$ at pH 4.5. These findings raise the possibility that these peroxidases have redundant function in vivo, possibly explaining the lack of attenuation we observed in Δoperon mutants in BALB/C mice at 35 days post infection. It is also possible that Δoperon mutants are not attenuated at this timepoint in BALB/C mice as the Tn-seq screen that previously identified Cfp29 as being required for growth in mouse lungs used the C57BL/6 strain (*Zhang et al., 2013*). KatG mutants are highly prevalent in the clinic as loss of function in KatG prevents activation of the INH prodrug. Importantly, many KatG mutants that fail to activate INH are also defective for catalase activity (*DeVito and Morris, 2003*; *Mo et al., 2004*; *Wei et al., 2003*). This raises the possibility that the DyP nanocompartment system could partially compensate for a loss of KatG catalase activity, enabling these drug resistant mutants to survive in vivo. In addition to a potential role for DyP in the context of INH treatment and INH resistant infections, we found that DyP encapsulin mutants are susceptible to treatment with PZA, an antibiotic known to require acidic conditions for efficacy. Thus, we predict that this encapsulin system may limit the efficacy of PZA treatment of patients.

In summary, we have demonstrated that Mtb produces a functional nanocompartment containing the peroxidase DyP. This system is essential for resisting oxidative stress at low pH, conditions that resemble the host lysosome. In addition, the DyP encapsulin system is required for growth in host macrophages and for resistance to the antibiotic PZA. Previously encapsulin systems have been primarily studied in the context of heterologous expression in *E. coli* and for bioengineering or structural studies. Our findings demonstrate the significance of encapsulin nanocompartments in bacterial physiology in the globally significant pathogen Mtb.

# Materials and methods

**Key resources table**

| Reagent type (species) or resource | Designation | Source or reference | Identifiers | Additional information |
|---|---|---|---|---|
| Gene (*Mycobacterium tuberculosis*) | Dyp | GenBank | Gene ID: 885388, Rv0799c | |
| Gene (*My. tuberculosis*) | Cfp29 | GenBank | Gene ID: 885460, Rv0798c | |
| Strain, strain background (*Mus musculus*) | BALB/C | The Jackson Laboratory | Stock no: 000651 | |
| Strain, strain background (*M. tuberculosis*) | H37Rv | Eric Rubin Lab, Harvard School of Public Health | | |
| Strain, strain background (*Escherichia coli*) | BL21 (DE3) LOBSTR | kerafast | Cat# EC1002 | |
| Genetic reagent (*M. tuberculosis*) | DyP::Tn | Broad Institute, Hung Lab | *Rv0799c::Tn* | |
| Genetic reagent (*M. tuberculosis*) | Δoperon | This study | *ΔRv0799c-Rv0798c* | See Materials and methods |

*Continued on next page*

*Continued*

| Reagent type (species) or resource | Designation | Source or reference | Identifiers | Additional information |
|---|---|---|---|---|
| Genetic reagent (*M. tuberculosis*) | ΔCfp29 | This study | Δ*Rv0798c* | See Materials and methods |
| Genetic reagent (*M. tuberculosis*) | ΔDyP | This study | Δ*Rv0799c* | See Materials and methods |
| Recombinant DNA reagent | Pet14b | Novagen | Cat# 69660-3 | |
| Recombinant DNA reagent | pUV15tetORm | Addgene | Cat# 17975 | AHT-inducible construct for all complementation |
| Recombinant DNA reagent | pKL4 | This study | | Rv0798c cloned into pUV15tetORm (with KanR) |
| Recombinant DNA reagent | pKL5 | This study | | Rv0799c cloned into pUV15tetORm (with KanR) |
| Recombinant DNA reagent | pKL6 | This study | | Operon cloned into pUV15tetORm (with KanR) |
| Recombinant DNA reagent | pKL14 | This study | | Operon cloned into pUV15tetORm (with HygR) |
| Recombinant DNA reagent | pKL15 | This study | | Rv0798c cloned into pUV15tetORm (with HygR) |
| Recombinant DNA reagent | pKL16 | This study | | Rv0799c cloned into pUV15tetORm (with HygR) |
| Recombinant DNA reagent | pUV15 pHGFP HygR: | Addgene | Cat# 70045 | Rv0799c cloned into pUV15tetORm (with HygR) |
| Recombinant DNA reagent | pMV762-mrx1-roGFP2 | Amit Singh, ICGEB, India | | PMC3907381 |
| Antibody | Anti-Mtb Cfp29 | Rabbit polyclonal | Produced by GenScript USA, see Materials and methods | 1:10,000 |
| Antibody | HRP | Goat anti-rabbit polyclonal | Santa Cruz Biotechnology sc-2030 | 1:5000 |
| Chemical compound, drug | 3-Ethylbenzothia zoline-6-sulfonic acid | Millipore Sigma | Cat# 10102946001 | |

## *M. tuberculosis* bacterial strains and plasmids

The Mtb strain H37Rv was used for all experiments. The transposon mutants DyP::Tn and Rv1762c::Tn were picked from an arrayed transposon mutant library generated at the Broad Institute. The ΔOperon and ΔCfp29 strains were made by homologous recombination using the pMSG361 vector (*Rosenberg et al., 2015*). For genetic complementation studies, the region encoding GFP and KanR in pUV15tetORm (*Ehrt et al., 2005*) was substituted via GoldenGate cloning with open reading frames for Rv0798c, Rv0799c, or the whole nanocompartment operon (Rv0798-99c). Expression of the complementation constructs was induced with anhydrotetracycline (200 ng/mL). *Dyp::Tn* mutants

were complemented with pKL14, 15, and 16; clean deletions (ΔOperon and ΔCfp29) were complemented with pKL4, 5, and 6. To measure redox homeostasis, strains were transformed with pMV762-mrx1-roGFP2 (*Bhaskar et al., 2014*). To measure intrabacterial pH, strains were transformed with pUV15-pHGFP (Addgene). The transposon mutant library for Tn-seq was generated in Mtb using the ΦMycoMarT7 transposon donor plasmid.

## *M. tuberculosis* bacterial cell culture

For infections, Mtb was grown to mid-log phase ($OD_{600}$ = 0.5–1.0) in Middlebrook 7H9 liquid medium supplemented with 10% albumin-dextrose-saline, 0.4% glycerol, and 0.05% Tween-80 or on solid 7H10 agar plates supplemented with Middlebrook OADC (BD Biosciences) and 0.4% glycerol. When specified, Tween-80 was substituted with 0.05% tyloxapol, and 10% albumin-dextrose-saline was prepared with fatty acid-free BSA (Sigma-Aldrich). Sauton's media was prepared with tyloxapol as previously specified (*Larsen et al., 2007*).

## DyP activity assays

Activity of the encapsulated and unencapsulated DyP was performed using methods adapted from *Contreras et al., 2014*. Briefly, DyP concentration for the encapsulated and unencapsulated DyP was determined by absorbance of the heme prosthetic group at 411 nm. Reactions were performed using 5 nM DyP, 480 nM $H_2O_2$, and 480 nM 2,2'-azino-bis (3-ethylbenzothiazoline-6-sulfonic acid) (ABTS) in 100 mM sodium citrate buffer pH 4–6. Product formation was monitored over 20 min via absorbance at 420 nm using a Varian Cary 50 UV-Vis Spectrophotometer (Agilent).

## Nanocompartment purification from Mtb

For each purification, 1.5 L of Mtb was grown to mid-log phase in standard 7H9 and washed with PBS. Bacteria were pelleted and lysed in buffer by bead beating (for 50 mL of buffer, PBS with 1 mM phenylmethylsulfonyl fluoride (PMSF) was supplemented with 50 mg lysozyme, 20 U DNaseI, and 100 µg RNase A). Lysates were passaged twice through 0.2 µm filters before removal from the BSL3. Clarified lysates were prepared by centrifugation at 20,000 × *g* for 20 min in a JA-20 rotor. Following clarification, 4–5 mL of lysate was layered onto top of 16 mL of 38% sucrose and centrifuged for 18 hr at 100,000 × *g* in a type 50.2 Ti rotor. The supernatant was discarded and the pellet was resuspended in 200 µL of PBS. Resuspended pellets were layered on top of a 10–50% discontinuous sucrose gradient and centrifuged for 21 hr at 100,000 × *g* in a SW 41 Ti rotor. The gradient was fractionated and aliquots from each fraction were analyzed by SDS-PAGE for the presence of Cfp29.

## Expression of holo-nanocompartment and naked DyP in *E. coli*

Plasmids for the expression of the holo-nanocompartment (DyP-loaded) and naked DyP constructs were designing using Gibson Assembly (NEB). Each construct was cloned into a pET-14-based destination vector containing a T7 promoter. The naked DyP construct contained an N-terminal poly-histidine tag for affinity purification. These constructs were transformed into *E. coli* BL21 (DE3) LOBSTR cells for protein overexpression. Cells were grown in LB media containing 60 µg/mL kanamycin at 37°C with shaking at 250 rpm until cultures reached an optical density ($OD_{600}$ = 0.5–0.6). Samples were then induced with 0.5 mM IPTG and grown overnight at 18°C. Liquid cultures were harvested by centrifugation at 5000 × *g* for 20 min at 4°C, flash frozen in liquid nitrogen, and then stored at –80°C for future use.

## Purification of holo-nanocompartment complex from *E. coli*

Cell pellets (5 g dry cell mass) were thawed at room temperature and resuspended in 50 mL of lysis buffer (20 mM Tris-HCl pH 8, 150 mM $NH_4Cl$, 20 mM $MgCl_2$) supplemented with 50 mg lysozyme, 20 U DNaseI, 100 µg RNase A. Samples were lysed by three passages through an Avestin EmulsiFlex-C3 homogenizer and clarified via centrifugation (15,000 × *g*, 30 min, 4°C). The clarified lysate was then spun at 110,000 × *g* for 3 hr at 4°C. The supernatant was discarded and the resulting pellet was resuspended with wash buffer (20 mM Tris pH 8, 150 mM $NH_4Cl$, 20 mM $MgCl_2$) supplemented with 1X Cell Lytic B (Sigma-Aldrich). The sample was then spun at 4000 × *g* at 4°C for 10 min followed by removing the supernatant and resuspension of the pellet in 4 mL of 50 mM Tris-HCl pH 8, 300 mM NaCl. The

sample was then incubated at room temperature for 10 min to allow for solubilization and then centrifuged at $4000 \times g$ at 4°C for 10 min to remove insoluble material. The resulting supernatant was then concentrated using Vivaspin 6 100000 MWCO concentrator columns (Sartorius). The sample was then purified via size-exclusion chromatography using a Superose 6 Increase column (GE Life Sciences), and fractions were analyzed by SDS-PAGE using 4–20% Criterion polyacrylamide gels (Bio-Rad) and visualized with GelCode Blue stain (Thermo Fisher).

## Purification of unencapsulated DyP from *E. coli*

Cell pellets (5 g dry cell mass) were thawed at room temperature and resuspended in 50 mL of buffer A (25 mM Tris HCl pH 7.5, 150 mM NaCl, 20 mM imidazole) supplemented with 50 mg lysozyme, 20 U DNaseI, 100 µg RNase A. Samples were lysed by three passages through an Avestin EmulsiFlex-C3 homogenizer and clarified via centrifugation ($15,000 \times g$, 30 min, 4°C). The resulting supernatant was then bound to HisPur Ni-NTA resin (Thermo Fisher Scientific) for 90 min at 4°C and then applied to a gravity column. The nickel resin was then washed with 30 resin volumes of buffer B (25 mM Tris-HCl pH 7.5, 150 mM NaCl, 40 mM imidazole) prior to eluting with buffer C (25 mM Tris-HCl pH 7.5, 150 mM NaCl, 350 mM imidazole). The eluate was then concentrated using Vivaspin 20 10000 MWCO concentrator columns (Sartorius) and desalted into 25 mM Tris pH 8, 300 mM NaCl using Econo-Pac10DG desalting columns (Bio-Rad). The SUMO tag was removed upon addition of SUMO protease at a 1:300 (SUMO protease:DyP) molar ratio and incubating overnight at 4°C. Purification was finished by size-exclusion chromatography with a Superose 6 Increase column (GE Life Sciences).

## Negative stain transmission electron microscopy

Nanocompartment samples were diluted to 50 nM and applied to Formvar/carbon-coated copper grids. The grids were then washed with MilliQ water three times followed by staining with 2% (w/v) uranyl acetate. Grids were examined using the FEI Tecnai 12, 120 kV transmission electron microscope, and images were captured with a charge-coupled device (CCD) camera.

## Exposure to oxidative and pH stress

Mtb was grown to mid-log phase in 7H9 media. Bacteria were diluted to $OD_{600} = 0.1$ in 10 mL of specified media at pH 4.5–6.5 and $H_2O_2$ was added to bacterial cultures at specified concentrations. Bacteria were incubated with stressors for 24 or 72 hr. CFUs were enumerated by diluting bacteria in PBS with 0.05% Tween-80 and plating serial dilutions on 7H10 agar.

## Measurement of redox homeostasis

Mtb strains were transformed with a plasmid expressing mrx1-roGFP2 and grown to mid-log phase in 7H9. Bacteria were diluted to $OD_{600} = 0.25$ in 200 µL of specified media and added to 96-well plates. Upon addition of $H_2O_2$ (5 mM), fluorescent emissions were recorded at 510 nm after excitation at 390 nm and 490 nm using a Spectramax M3 spectrophotometer. Values reported are emissions ratios (390 nm/490 nm) and were measured 60 min following addition of $H_2O_2$.

## Measurement of intrabacterial pH

Mtb strains were transformed with a plasmid expressing pHGFP and grown to mid-log phase in 7H9. To prepare standards, $1.5 \times 10^8$ bacterial cells were pelleted and resuspended in 400 µL lysis buffer (50 mM Tris-HCl pH 7.5, 5 mM EDTA, 0.6% sodium dodecyl sulfate (SDS), 1 mM PMSF) before bead beating. Cell debris were pelleted and clarified lysates were kept at 4°C until use, at which point 10 µL of clarified lysate were added to 200 µL of medium with varying pH levels (4.5–8.0). To prepare samples, $1.5 \times 10^8$ bacterial cells were pelleted and washed with PBS twice before being resuspended in specified media and diluted to $OD_{600} = 0.5$ in 200 µL of medium and added to 96-well plates. Upon addition of $H_2O_2$ (5 mM), fluorescent emissions were recorded at 510 nm following excitation at 395 nm and 475 nm. Values reported were interpolated from 395/475 ratios obtained from the standard curve.

## Western blot analysis of Cfp29 expression

Mtb strains were grown to mid-log phase in 7H9 medium. Bacteria were pelleted and washed twice with PBS prior to resuspension in lysis buffer (50 mM Tris-HCl pH 7.5, 5 mM EDTA, 0.6% SDS, 1 mM

PMSF). Samples were lysed using a bead-beater, and cell debris were pelleted. Clarified lysates were heat-sterilized at 100°C for 15 min and frozen prior to use. Total protein lysates were analyzed by SDS-PAGE using precast Tris-HCl 4–20% criterion gels (Bio-Rad). Primary polyclonal antibodies for Cfp29 were generated by GenScript USA Inc via immunization of rabbits with three peptides from the protein sequence. HRP-conjugated goat anti-rabbit IgG secondary antibodies were used (sc-2030; Santa Cruz Biotechnology). Western Lightning Plus-ECL chemiluminescence substrate (Perkin Elmer) was used, and blots were developed using a ChemiDoc MP System (Bio-Rad).

## Infection of murine macrophages

Macrophages were derived from bone marrow of C57BL/6 mice by flushing cells from femurs. Cells were cultured in Dulbecco's Modified Eagle Medium (DMEM) supplemented with 10% fetal bovine serum (FBS) and 10% supernatant from 3T3-M-CSF cells for 6 days, with feeding on day 3. After differentiation, bone marrow-derived macrophages (BMDMs) continued to be cultured in BMDM media containing M-CSF. For infection, BMDMs were seeded at a density of $5 \times 10^4$ cells per well in a 96-well dish. BMDMs were allowed to adhere overnight and then infected with DMEM supplemented with 5% FBS and 5% horse serum at a multiplicity of infection (MOI) of 1. Following a 4 hr phagocytosis period, infection medium was removed and cells were washed with room temperature PBS before fresh medium was added. For CFU enumeration, medium was removed and cells were lysed in water with 0.5% Triton-X and incubated at 37°C for 10 min. Following the incubation, lysed cells were resuspended and serially diluted in PBS with 0.05% Tween-80. Dilutions were plated on 7H10 plates.

## Infection of mice

BALB/C mice were obtained from the Jackson Laboratory, Bar Harbor, ME. Mice were infected at 6 weeks of age with 250 CFUs of Mtb strains by the aerosol route using a Glas-Col (Brazil, IN) full-body inhalation exposure system. Infections were allowed to proceed for 35 days at which time mice were euthanized and CFU from the lungs enumerated by plating on 7H10 plates. PZA treatment began 14 days after infection. PZA (Acros Organics) was formulated in water and administered 5 days per week, once per day, at 150 mg/kg, by oral gavage. We used G*power to calculate the needed number of animals to power the mouse study and used five mice per group.

## Transposon-sequencing screen

A transposon mutant library in H37Rv was grown to mid-log phase in 7H9. Bacteria were diluted to $OD_{600}$ = 0.1 in 10 mL 7H9 at pH 4.5 with 2.5 mM $H_2O_2$. Mutants were exposed to these stressors for 72 hr and then diluted to 15,000 CFU/mL in PBS with 0.05% Tween-80. Approximately 30,000 bacteria were plated onto six 245 mm × 245 mm 7H10 plates supplemented with 0.05% Tween-80 and kanamycin (50 µg/mL). Control libraries were not exposed to low pH or $H_2O_2$ and were plated onto 7H10 plates. Colonies grew for 21 days and were collected for genomic DNA isolation. Samples for sequencing were prepared by the University of California, Davis Genome Center DNA Technologies Core, by following the protocol outlined by *Long et al., 2015*. PE100 reads were run on an Illumina HiSeq with ~20 million reads per sample. Sample alignment and TRANSIT preprocessing were performed by the University of California, Davis Bioinformatics Group, as previously outlined (*DeJesus et al., 2015*). TRANSIT analysis was performed as specified by *DeJesus et al., 2015*. Resampling analysis was performed using the reference genome H37RvBD_prot and the following parameters: for global options, 0% of the N- and C- terminus were ignored; for resampling options, 10,000 samples were taken and normalized using the TTR function. Correction for genome positional bias was performed. Statistical significance was determined by p-value ≤ 0.05 and log2 fold change ≤–1 or by p-adjusted value ≤0.05.

## Acknowledgements

We thank Tom Ioerger and Michael DeJesus for assistance with TRANSIT analysis and Amit Singh for the kind gift of pMV762-mrx1-roGFP2. We thank Jeff Cox and members of the Cox lab for helpful discussions. This work was supported by funding from the Center for Emerging and Neglected disease for funding to KAL, the National Institute of General Medical Sciences (R01GM129241) to DFS, and the National Institute of Allergy and Infectious Diseases (1R01AI143722) to SAS.

## Additional information

### Funding

| Funder | Grant reference number | Author |
| --- | --- | --- |
| University of California, Berkeley | Graduate Student Award | Katie A Lien |
| National Institute of General Medical Sciences | R01GM129241 | David F Savage |
| National Institute of Allergy and Infectious Diseases | 1R01AI143722 | Sarah A Stanley |

The funders had no role in study design, data collection and interpretation, or the decision to submit the work for publication.

### Author contributions

Katie A Lien, Conceptualization, Data curation, Formal analysis, Investigation, Methodology, Validation, Writing – original draft, Writing – review and editing; Kayla Dinshaw, Data curation, Formal analysis, Investigation, Writing – original draft, Writing – review and editing; Robert J Nichols, Formal analysis, Investigation, Methodology, Writing – review and editing; Caleb Cassidy-Amstutz, Formal analysis, Investigation, Methodology; Matthew Knight, Investigation; Rahul Singh, Methodology, Resources, Writing – review and editing; Lindsay D Eltis, Conceptualization, Methodology, Resources, Writing – review and editing; David F Savage, Conceptualization, Investigation, Methodology, Supervision, Writing – review and editing; Sarah A Stanley, Conceptualization, Data curation, Formal analysis, Funding acquisition, Methodology, Project administration, Supervision, Writing – original draft, Writing – review and editing

### Author ORCIDs

Katie A Lien ![ORCID] http://orcid.org/0000-0003-2721-0887
Kayla Dinshaw ![ORCID] http://orcid.org/0000-0002-5354-746X
Robert J Nichols ![ORCID] http://orcid.org/0000-0002-8476-0554
David F Savage ![ORCID] http://orcid.org/0000-0003-0042-2257
Sarah A Stanley ![ORCID] http://orcid.org/0000-0002-4182-9048

### Ethics

All procedures involving the use of mice were approved by the University of California, Berkeley Institutional Animal Care and Use Committee (protocol no. R353-1113B). All protocols conform to federal regulations, the National Research Council's Guide for the Care and Use of Laboratory Animals, and the Public Health Service's Policy on Humane Care and Use of Laboratory Animals.

### Decision letter and Author response

Decision letter https://doi.org/10.7554/eLife.74358.sa1
Author response https://doi.org/10.7554/eLife.74358.sa2

## Additional files

### Supplementary files

- Transparent reporting form
- Supplementary file 1. Mtb transposon sequencing screen in acidified broth with perxoide stress.
- Source data 1. Original scanned image of the gel depicting *Figure 1—figure supplement 1A*.
- Source data 2. Scanned image of the gel depicting *Figure 1—figure supplement 1A*.
- Source data 3. Original scanned image of the gel depicting *Figure 1—figure supplement 1B*.
- Source data 4. Original scanned image of the gel depicting *Figure 1—figure supplement 1B*.
- Source data 5. Original scanned image of the gel depicting *Figure 1—figure supplement 1C*.
- Source data 6. Original image capture of PVDF membrane to show the molecular weight ladder (*Figure 1—figure supplement 1D*).

• Source data 7. Original fluorescence image of western blot with Ag85B antibody (*Figure 1—figure supplement 1D*).

• Source data 8. Original fluorescence image of western blot with Cfp29 antibody (*Figure 1—figure supplement 1D*).

• Source data 9. Western images from *Figure 1—figure supplement 1D*.

• Source data 10. Original scanned image of the gel depicting both Tn::DypB+DypB and Tn::DypB+Cfp29 (*Figure 1—figure supplement 1E*).

• Source data 11. Original scanned image of the gel depicting Tn::DypB+Operon (*Figure 1—figure supplement 1E*).

• Source data 12. This image depicts Tn::DypB+DypB, Tn::DypB+Cfp29, and Tn::DypB+Operon with molecular weight markers, lane labels, and the section of the image that is depicted in the main figure outlined in blue (*Figure 1—figure supplement 1E*).

• Source data 13. Wig file for control (library only) replicate 1 (*Figure 3A and B*).

• Source data 14. Wig file for control (library only) replicate 2 (*Figure 3A and B*).

• Source data 15. Wig file for control (library only) replicate 3 (*Figure 3A and B*).

• Source data 16. Wig file for experimental ($H_2O_2$_low pH) replicate 1 (*Figure 3A and B*).

• Source data 17. Wig file for experimental ($H_2O_2$_low pH) replicate 2 (*Figure 3A and B*).

• Source data 18. Wig file for experimental ($H_2O_2$_low pH) replicate 3 (*Figure 3A and B*).

## Data availability
All data generated or analysed during this study are included in the manuscript and supporting file.

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
