## [Editor Report]

The authors identify a two-gene operon encoding for a peroxidase and an encapsulin in *M. tuberculosis*. They demonstrate that this system is important for *M. tuberculosis* to resist oxidative stress at acidic pH in vitro and in vivo in infected macrophages. The work comprises a significant advance in the understanding of mycobacterial pathogenicity and the role of bacterial encapsulins.

---

## [Decision Letter]

[Editors’ note: the authors submitted for reconsideration following the decision after peer review. What follows is the decision letter after the first round of review.]

Thank you for submitting your work entitled "A nanocompartment containing the peroxidase DypB contributes to defense against oxidative stress in *M. tuberculosis*" for consideration by *eLife*. Your article has been reviewed by 3 peer reviewers, and the evaluation has been overseen by a Reviewing Editor and a Senior Editor. The following individuals involved in review of your submission have agreed to reveal their identity: Neeraj Dhar (Reviewer #2).

Our decision has been reached after consultation between the reviewers. Based on these discussions and the individual reviews below, we regret to inform you that your work will not be considered further for publication in *eLife*.

As you will see from the detailed reviewer comments, there are several interesting aspects of the work but at this stage, there are notable limitations also. Given this, we are rejecting the manuscript in its current form. That said, your findings are compelling indeed and if you are able to address the concerns, I would be willing to reconsider the study as a new manuscript, where you indicate how you have dealt with the reviewer's comments from the first round of review.

*Reviewer #1:*

Lien and colleagues report on the role of encapsulin Cfp29/Rv0798c and peroxidase DypB/Rv0799c in the resistance of *M. tuberculosis* to oxidative stress. The study builds on a previous study (Contreras et al. 2014) which reported the detection of Cfp29-encapsulated DypB in recombinant *E. coli*; here, the authors show that encapsulation also occurs in *M. tuberculosis*. In addition, the authors show that a cfp29/dypB-null mutant is highly sensitive to H202 at acidic pH, and their data suggest that encapsulation of DypB increases peroxidase activity. Finally, the authors report that dypB- and cfp29/dypB-null mutants are altered in their ability to multiply within macrophages.

Overall, the concept of compartmentalization in prokaryotes is of great interest; perhaps the most interesting conclusion is that the encapsulation of a virulence factor contributes to resistance to host-imposed stress in a major pathogen, although the role of Cfp29 in enhancing DypB activity seems rather modest, both in vitro and in vivo. However, the study as a whole contains many results that are poorly related (see comments below), the reader is finally left with an attractive hypothesis that is not sufficiently explored mechanistically, and the study appears a bit preliminary in this respect. How are nanocapsules secreted? Are they present in the mycobacterial vacuole? Is the phenotype of dypB- and cfp29/dypB-null mutants in macrophages really related to oxidative stress? What is the phenotype of these mutants in vivo? Addressing these questions, and perhaps others, would have allowed a more comprehensive study, instead of including unrelated or poorly related data on KatG, lipids, etc.

– Data suggesting that "DypB encapsulation enhances protection against oxidative stress under low pH conditions" are not entirely convincing. In Figure 2D, the expression of DypB in the DypB::Tn mutant almost fully restores resistance to H_2_O_2_. Is the difference between DypB::Tn+pDypB and DypB::Tn+pOperon significant? Similarly, in Figure 2E, the phenotype of the cfp29-null mutant should be restored by genetic complementation.

– The phenotype of the Rv1762c mutant is quite clear (Figure 2F), but the mutant has not been complemented, and more importantly, the question of whether this is related to encapsulation is not explored at all; I suggest deleting this data as it adds nothing to the overall message.

– "Based on the Tn-seq data, we hypothesized that lipids may be responsible for the sensitivity of DypB mutants to oxidative stress under acidic conditions". I do not understand this hypothesis. The Tn-seq data reveal mutants that are attenuated under conditions of acidic and oxidative stress. As far as I understand, the experiment was not designed to identify the genes involved in the sensitivity of the DypB mutant, which would have required a library of Tn mutants in a dypB-null background.

– The effect of lipids on the phenotype of the operon-null mutant is clear; but the reader is given no further explanation here. The same applies for katG/INH; these results are poorly related to the study as a whole.

– The phenotype of both mutants in macrophages is quite modest and should be verified by genetic complementation.

*Reviewer #2:*

In this manuscript the authors describe and characterize the role of a two-gene operon predicted to encode for an encapsulated nanocompartment in *M. tuberculosis*. The authors show that heterologous expression of these genes in *E. coli* produces Cfp29 nanocompartments that encapsulate the DypB peroxidase and protect it from degradation. They also provide first evidence that Cfp29-DypB nanocompartments are produced by Mtb under normal growth conditions. Further characterization revealed that DypB had maximal activity at low pH, an environment that *M. tuberculosis* faces during its intracellular lifestyle. To determine if the encapsulated DypB serves a protective role against oxidative stress, M. tb strains carrying deletions of the DypB operon were tested against oxidative stress under normal and acidic conditions. The mutant strain was found to be more sensitive to oxidative stress especially in acidic environments. A transposon sequencing screening identified DypB and KatG as the only peroxidades required for survival to oxidative stress at low pH, together with other genes involved in lipid or cholesterol metabolism. Interestingly, the hypersusceptibility of the DypB mutant was found to be dependent on free lipids present in the culture medium. Use of a redox potential fluorescent reporter revealed the mutant to have a more oxidizing intracellular environment compared to wild-type bacteria. These observations leaded to the conclusion that DypB nanocompartments protect bacteria from oxidative stress, low pH, and fatty acids suggesting that this system may be important for survival in phagosomes. This hypothesis is proved by infecting bone-marrow derived macrophages with the WT and the double mutant strains and showing that the mutant is attenuated. Finally, the double mutant was also shown to be more susceptible to Pyrazinamide suggesting that nanocompartments are important for resistance to oxidative stress.

The manuscript is written well and the experiments have been planned and executed well, to demonstrate the role of the DypB peroxidase against oxidative stress.

The only major concern is the relatively lesser focus on the encapsulation aspect. The data showing the presence of nanocompartments in Mtb and the encapsulation of DypB is quite convincing. However, all the subsequent characterization and most of the follow-up experiments are done with the mutant strain lacking both the DypB as well as the cfp29 genes. The authors already have a strain (∆cfp29) that expresses an unencapsulated DypB ( or even DypB::tn+pDypB). These mutants should be included in the stress-susceptibility experiments shown in Figures 3 and 4. This set of experiments along with the ones with the deletion of the complete operon will allow the clearer understanding of the role of DypB and the process of encapsulation. Since the main focus of the manuscript is about nanocompartments, disentangling this is crucial to the message.

*Reviewer #3:*

In Lien et al., the authors investigate a nanocompartment in the human pathogen *Mycobacterium tuberculosis* and explore how the cargo DypB, a peroxidase, contributes to bacterial defense against oxidative and acidic stress. Towards this, the authors have isolated nanocompartments and identified DypB as cargo. They show that mutants lacking the nanocompartment are attenuated in vitro by H_2_O_2_ in an acidic environment in the presence of lipids. The authors propose an attractive hypothesis-- that the bacteria encapsulate DypB to prevent its degradation by host cellular proteases and that it plays a role in resisting oxidative and acidic stress that the bacteria encounter inside macrophages. They show that DypB and nanocompartment mutants are attenuated in their survival inside the macrophages, although they do not show that this is due to their enhanced susceptibility to pH and oxidative stress. Overall, this is an interesting manuscript which has the potential to add important insight into encapsulation systems and the defense mechanisms that *M. tuberculosis* employs to survive inside acid and ROS rich phagosomes. The work, however, seems preliminary. It could be substantially strengthened by additional macrophage experiments, which currently lack important controls and are under-developed, as well as in vivo experiments.

1. The macrophage experiments need to include the complemented strains.

2. The macrophage experiments should include conditions that alter ROS, such as IFN-γ treatment, Nox2 knockouts, ROS neutralizing agents. It would also be informative to include pH and/or cathepsin neutralizing treatments. The Mrrx-GFP reporter could also be used to assess the redox state of the bacteria in macrophages.

3. The authors should determine whether the encapsulation mutant is attenuated in macrophages.

4. What contribution does DypB and the encapsulation system make to virulence in vivo?

5. It remains unclear how nanocompartments get to the extracellular space or if that is where they function. It would be useful to determine whether purified nanocompartments can rescue the mutant in vitro and in macrophages.

[Editors’ note: further revisions were suggested prior to acceptance, as described below.]

Thank you for submitting your article "A nanocompartment system contributes to defense against oxidative stress in *M. tuberculosis*" for consideration by *eLife*. Your article has been reviewed by 2 peer reviewers, and the evaluation has been overseen by a Reviewing Editor and Bavesh Kana as the Senior Editor. The reviewers have opted to remain anonymous.

Essential revisions:

1. Statistics should be provided in Figure 3E.

2. There seems to be a mis-annotation in Figure 4B.

3. Complementation in Figure 4A,B was difficult and the authors should clearly state their arguments in the manuscript (as they did in the reply to reviewers comment), so as not to leave a gap in this regard.

4. Legend to Figure 2: Initially it is mentioned that in panels C, E and F, the bacteria were exposed to 2. 5 mM H_2_O_2_ for 72 hours. Later it is mentioned that in D, E, F they were exposed for 24 hours. In the text it is mentioned that the bacteria were exposed for 72h. Clarify which one is accurate. Also, panel F is listed as representative of 2 or 3 experiments. Which one is correct? In panel 2C, the fourth bar is labelled as DypB. Should be DyP. For consistency, please plot y-axis in 2F in log scale.

---

## [Author Response]

[Editors’ note: the authors resubmitted a revised version of the paper for consideration. What follows is the authors’ response to the first round of review.]

Reviewer #1:Lien and colleagues report on the role of encapsulin Cfp29/Rv0798c and peroxidase DypB/Rv0799c in the resistance of M. tuberculosis to oxidative stress. The study builds on a previous study (Contreras et al. 2014) which reported the detection of Cfp29-encapsulated DypB in recombinant *E. coli*; here, the authors show that encapsulation also occurs in M. tuberculosis. In addition, the authors show that a cfp29/dypB-null mutant is highly sensitive to H202 at acidic pH, and their data suggest that encapsulation of DypB increases peroxidase activity. Finally, the authors report that dypB- and cfp29/dypB-null mutants are altered in their ability to multiply within macrophages.Overall, the concept of compartmentalization in prokaryotes is of great interest; perhaps the most interesting conclusion is that the encapsulation of a virulence factor contributes to resistance to host-imposed stress in a major pathogen, although the role of Cfp29 in enhancing DypB activity seems rather modest, both in vitro and in vivo. However, the study as a whole contains many results that are poorly related (see comments below), the reader is finally left with an attractive hypothesis that is not sufficiently explored mechanistically, and the study appears a bit preliminary in this respect. How are nanocapsules secreted? Are they present in the mycobacterial vacuole? Is the phenotype of dypB- and cfp29/dypB-null mutants in macrophages really related to oxidative stress? What is the phenotype of these mutants in vivo? Addressing these questions, and perhaps others, would have allowed a more comprehensive study, instead of including unrelated or poorly related data on KatG, lipids, etc.– Data suggesting that "DypB encapsulation enhances protection against oxidative stress under low pH conditions" are not entirely convincing. In Figure 2D, the expression of DypB in the DypB::Tn mutant almost fully restores resistance to H_2_O_2_. Is the difference between DypB::Tn+pDypB and DypB::Tn+pOperon significant? Similarly, in Figure 2E, the phenotype of the cfp29-null mutant should be restored by genetic complementation.

The reviewer correctly points out that DypB accounts for the majority of the phenotype when TB is exposed to H_2_O_2_ at low pH. However, we do observe that Cfp29 also makes a smaller, yet reproducible and statistically significant contribution. To make this more convincing we have added the significance analysis to figure 2D. We have also complemented the Cfp29 mutant as requested, new Figure 2F. Given that there is a lack of any demonstration in the literature that an encapsulin protein is important in the physiology of a bacterium, despite the widespread presence of encapsulins in hundreds of bacterial species, we believe that this result, though somewhat modest, is important.

– The phenotype of the Rv1762c mutant is quite clear (Figure 2F), but the mutant has not been complemented, and more importantly, the question of whether this is related to encapsulation is not explored at all; I suggest deleting this data as it adds nothing to the overall message.

We thank the reviewer for pointing this out and have removed the data as suggested.

– "Based on the Tn-seq data, we hypothesized that lipids may be responsible for the sensitivity of DypB mutants to oxidative stress under acidic conditions". I do not understand this hypothesis. The Tn-seq data reveal mutants that are attenuated under conditions of acidic and oxidative stress. As far as I understand, the experiment was not designed to identify the genes involved in the sensitivity of the DypB mutant, which would have required a library of Tn mutants in a dypB-null background.

The reviewer is correct that the Tn-seq experiment was conducted to identify peroxidases required under conditions of oxidative stress at low pH. However, because Tn-Seq is comprehensive, we also identified a wider set of genes required for resistance under these conditions. We were intrigued by our observation that mutants encoding multiple subunits of the Mce1 fatty acid importer were susceptible under these conditions. This led us to hypothesize that importing fatty acids might protect the bacteria from fatty acid toxicity generated under oxidative stress at low pH. This caused us to test whether eliminating fatty acids from the medium entirely eliminated the effect, which we did observe. We realize we did not explain this rationale clearly in the text, so we thank the reviewer for raising this question. We have revised the text to be more clear.

– The effect of lipids on the phenotype of the operon-null mutant is clear; but the reader is given no further explanation here. The same applies for katG/INH; these results are poorly related to the study as a whole.

Again, we thank the reviewer for pointing out insufficient explanation of the results we have in the text – we have added more discussion and context. However, we do acknowledge that we do not yet understand the role of fatty acids in the phenotype we describe. This is a recently discovered biological system about which almost nothing is known in any bacterial species, and there is much we need to learn moving forward. However, we would argue that it is beyond the scope of the first publication on the endogenous system in Mtb to fully elucidate the mechanism.

– The phenotype of both mutants in macrophages is quite modest and should be verified by genetic complementation.

The reviewer correctly points out that the phenotype is modest in macrophages. The phenotypes of TB bacterial mutants are often modest in macrophages, given the limited dynamic range of growth. We have attempted to complement these results however, complementation of the nanocompartment system has proven to be difficult, particularly in macrophages. The limited dynamic range between wild-type and mutant in macrophages also makes interpretation of the complementation results tricky.

To confirm that the phenotype of these mutants in macrophages is due to the loss of the two gene encapsuling operon, we performed this experiment with two independently derived mutants >5 times, and always observe attenuation of both mutants in macrophages.

Furthermore, these genes are predicted to be in a 2 gene operon with no other predicted genes in close proximity, therefore polar effects are extremely unlikely. We therefore stand by this data and would like to include it in the final manuscript, even though we have not observed attenuation in vivo*.* We believe that this is important data for other researchers in the field who may be interested in examining nanocompartments in other bacterial pathogens.

Reviewer #2:In this manuscript the authors describe and characterize the role of a two-gene operon predicted to encode for an encapsulated nanocompartment in M. tuberculosis. The authors show that heterologous expression of these genes in *E. coli* produces Cfp29 nanocompartments that encapsulate the DypB peroxidase and protect it from degradation. They also provide first evidence that Cfp29-DypB nanocompartments are produced by Mtb under normal growth conditions. Further characterization revealed that DypB had maximal activity at low pH, an environment that M. tuberculosis faces during its intracellular lifestyle. To determine if the encapsulated DypB serves a protective role against oxidative stress, M. tb strains carrying deletions of the DypB operon were tested against oxidative stress under normal and acidic conditions. The mutant strain was found to be more sensitive to oxidative stress especially in acidic environments. A transposon sequencing screening identified DypB and KatG as the only peroxidades required for survival to oxidative stress at low pH, together with other genes involved in lipid or cholesterol metabolism. Interestingly, the hypersusceptibility of the DypB mutant was found to be dependent on free lipids present in the culture medium. Use of a redox potential fluorescent reporter revealed the mutant to have a more oxidizing intracellular environment compared to wild-type bacteria. These observations leaded to the conclusion that DypB nanocompartments protect bacteria from oxidative stress, low pH, and fatty acids suggesting that this system may be important for survival in phagosomes. This hypothesis is proved by infecting bone-marrow derived macrophages with the WT and the double mutant strains and showing that the mutant is attenuated. Finally, the double mutant was also shown to be more susceptible to Pyrazinamide suggesting that nanocompartments are important for resistance to oxidative stress.The manuscript is written well and the experiments have been planned and executed well, to demonstrate the role of the DypB peroxidase against oxidative stress.The only major concern is the relatively lesser focus on the encapsulation aspect. The data showing the presence of nanocompartments in Mtb and the encapsulation of DypB is quite convincing. However, all the subsequent characterization and most of the follow-up experiments are done with the mutant strain lacking both the DypB as well as the cfp29 genes. The authors already have a strain (∆cfp29) that expresses an unencapsulated DypB ( or even DypB::tn+pDypB). These mutants should be included in the stress-susceptibility experiments shown in Figures 3 and 4. This set of experiments along with the ones with the deletion of the complete operon will allow the clearer understanding of the role of DypB and the process of encapsulation. Since the main focus of the manuscript is about nanocompartments, disentangling this is crucial to the message.

We agree with this point and have performed additional experiments with the Cfp29 mutant. First, we have shown that although the DyP mutant has defects in intracellular redox state at baseline, which is exacerbated under conditions of oxidative stress, we do not observe a change in redox state in a Cfp29 mutant (Figure 2G). We have added this to the text. In addition, we have not observed a phenotype of the Cfp29 mutant in exposure to PZA, however it is possible that is because we have not identified conditions under which the bacteria are most susceptible to this antibiotic. We do find consistently that the Cfp29 protein plays a small but significant role in protecting against exposure to oxidative stress at low pH, and we do believe that as we continue to explore this system we will better understand under which encapsulation is important for physiological function. It is important to note that we provide the first evidence that encapsulin systems contribute to oxidative stress in a bacterial pathogen, and only the second demonstration of a functional encapsulin system in any species. We therefore think these findings while exciting and informative for microbiologists across disciplines, are still by necessity preliminary due to the nascent status of the field.

We have expanded our discussion of the significance of these results in the Discussion section.

Reviewer #3:In Lien et al., the authors investigate a nanocompartment in the human pathogen Mycobacterium tuberculosis and explore how the cargo DypB, a peroxidase, contributes to bacterial defense against oxidative and acidic stress. Towards this, the authors have isolated nanocompartments and identified DypB as cargo. They show that mutants lacking the nanocompartment are attenuated in vitro by H_2_O_2_ in an acidic environment in the presence of lipids. The authors propose an attractive hypothesis-- that the bacteria encapsulate DypB to prevent its degradation by host cellular proteases and that it plays a role in resisting oxidative and acidic stress that the bacteria encounter inside macrophages. They show that DypB and nanocompartment mutants are attenuated in their survival inside the macrophages, although they do not show that this is due to their enhanced susceptibility to pH and oxidative stress. Overall, this is an interesting manuscript which has the potential to add important insight into encapsulation systems and the defense mechanisms that M. tuberculosis employs to survive inside acid and ROS rich phagosomes. The work, however, seems preliminary. It could be substantially strengthened by additional macrophage experiments, which currently lack important controls and are under-developed, as well as in vivo experiments.1. The macrophage experiments need to include the complemented strains.

Please see our comment to reviewer 1.

2. The macrophage experiments should include conditions that alter ROS, such as IFN-γ treatment, Nox2 knockouts, ROS neutralizing agents. It would also be informative to include pH and/or cathepsin neutralizing treatments. The Mrrx-GFP reporter could also be used to assess the redox state of the bacteria in macrophages.

We thank the reviewer for this suggestion. Although we have attempted numerous times to evaluate the phenotype of the mutants in *phox* deficient macrophages, we have observed inconsistent results. Although the mutant is always attenuated in wild-type macrophages, our results from the *phox* mutant are inconsistent. Similarly, we also attempted to measure the redox state of the bacteria in macrophages, but were unable to reproduce even the published data using this reporter in wild-type bacteria in macrophages, so have been unable to generate reliable data comparing wild-type and mutant bacteria.

3. The authors should determine whether the encapsulation mutant is attenuated in macrophages.

We have done these experiments, however due to the limited dynamic range of macrophage infections with Mtb we have inconsistent results. We have observed that the mutant is attenuated, but the differences are not statistically significant.

4. What contribution does DypB and the encapsulation system make to virulence in vivo?

We have performed an in vivo infection of BALB/C mice, due to the precedence for using this strain in studies of the antibiotic PZA, and do not see significant attenuation of the operon mutant. We are currently investigating whether attenuation of the mutant is mouse strain dependent (as Cfp29 was previously a hit in a transposon screen in C57BL/6 mice from the Rubin lab) and/or whether KatG can compensate for the loss of this system, however this will take some time to unravel and we feel is beyond the scope of the current manuscript.

5. It remains unclear how nanocompartments get to the extracellular space or if that is where they function. It would be useful to determine whether purified nanocompartments can rescue the mutant in vitro and in macrophages.

We have attempted these experiments and have seen a small effect of adding purified nanocompartments back in the in vitro system. However, we elected not to include this data as we feel it is difficult to interpret. First, it is difficult to purify sufficient material to achieve concentrations of nanocompartment that may be achieved within a phagosome. Second, we feel that it is difficult to extrapolate the biological significance of a positive result, as providing an enzyme that can function as a catalase should protect bacteria from oxidative stress, regardless of whether it may function naturally outside the cell. We do not feel that this is an important point of the story at this time, and for the sake of clarity prefer to omit the data.

[Editors’ note: what follows is the authors’ response to the second round of review.]

Essential revisions:1. Statistics should be provided in Figure 3E.

We have included statistics and thank the reviewers for pointing out this oversight.

2. There seems to be a mis-annotation in Figure 4B.

We have corrected the mis-annotated symbol.

3. Complementation in Figure 4A,B was difficult and the authors should clearly state their arguments in the manuscript (as they did in the reply to reviewers comment), so as not to leave a gap in this regard.

We have added an explanation to the text as requested.

4. Legend to Figure 2: Initially it is mentioned that in panels C, E and F, the bacteria were exposed to 2. 5 mM H_2_O_2_ for 72 hours. Later it is mentioned that in D, E, F they were exposed for 24 hours. In the text it is mentioned that the bacteria were exposed for 72h. Clarify which one is accurate. Also, panel F is listed as representative of 2 or 3 experiments. Which one is correct? In panel 2C, the fourth bar is labelled as DypB. Should be DyP. For consistency, please plot y-axis in 2F in log scale.

We apologize for the confusion and have clarified all of the requested information. The correct timepoint is 72h; 2F was done twice and 2G was done three times; we have changed DypB to Dyp; we have changed 2F to a log scale as requested.